# A cytokine panel and procalcitonin in COVID-19, a comparison between intensive care and non-intensive care patients

Tina Mazaheri[1]*, Ruvini Ranasinghe[1], Wiaam Al-Hasani[1], James Luxton[2], Jessica Kearney[3], Allison Manning[4], Georgios K. Dimitriadis[3,5], Tracey Mare[2], Royce P. Vincent[1,5]

1 Department of Clinical Biochemistry, King's College Hospital NHS Foundation Trust, London, United Kingdom, 2 Contract R&D Department (Viapath), King's College Hospital NHS Foundation Trust, London, United Kingdom, 3 Department of Endocrinology ASO/EASO COM, King's College Hospital NHS Foundation Trust, London, United Kingdom, 4 CAPA Intern (Clinical Biochemistry), King's College Hospital NHS Foundation Trust, London, United Kingdom, 5 Obesity, Type 2 Diabetes and Immunometabolism Research Group, Department of Diabetes, Faculty of Life Sciences, School of Life Course Sciences, King's College London, London, United Kingdom

* tina.mazaheri@nhs.net

**Data Availability Statement:** All relevant data are within the manuscript and its Supporting Information files.

## Abstract

### Objectives

Procalcitonin (PCT) is an acute-phase reactant with concentrations ≥0.5 μg/L indicative of possible bacterial infection in patients with SARS-CoV-2 infection (COVID-19). Some with severe COVID-19 develop cytokine storm secondary to virally driven hyper-inflammation. However, increased pro-inflammatory cytokines are also seen in bacterial sepsis. This study aimed to assess the clinical utility of a cytokine panel in the assessment of COVID-19 with bacterial superinfections along with PCT and C-reactive protein (CRP).

### Methods

The retrospective analysis included serum cytokines (interleukins; IL-1β, IL-6, IL-8 and tumour necrosis factor (TNFα)) measured using Ella™ (Bio-Techne, Oxford, UK) and PCT measured by Roche Cobas (Burgess Hill, UK) in patients admitted with COVID-19 between March 2020 and January 2021. Patients enrolled into COVID-19 clinical trials, treated with Remdesivir/IL-6 inhibitors were excluded. The cytokine data was compared between intensive care unit (ICU) patients, age matched non-ICU patients and healthy volunteers as well as ICU patients with high and normal PCT (≥0.5 vs. <0.5 μg/L).

### Results

Cytokine concentrations and CRP were higher in COVID-19 patients (76; ICU & non-ICU) vs. healthy controls (n = 24), all p<0.0001. IL-6, IL-8, TNFα and were higher in ICU patients (n = 46) vs. non-ICU patients (n = 30) despite similar CRP. Among 46 ICU patients, the high PCT group (n = 26) had higher TNFα (p<0.01) and longer ICU stay (mean 47 vs. 25 days,

**Funding:** The author(s) received no specific funding for this work.

**Competing interests:** The authors have declared that no competing interests exist.

p<0.05). There was no difference in CRP and blood/respiratory culture results between the groups.

## Conclusions

Pro-inflammatory cytokines and PCT were higher in COVID-19 patients requiring ICU admission vs. non-ICU admissions despite no difference in CRP. Furthermore, TNFα was higher in those with high PCT and requiring longer ICU admission despite no difference in CRP or rate of bacterial superinfection.

## Introduction

Coronavirus disease 2019 (COVID-19) pandemic caused by the Severe Acute Respiratory Syndrome Coronavirus 2 (SARS-CoV-2) continues to challenge medical centres across the world. Severe COVID-19 infection in some patients can activate excessive inflammatory response leading to overproduction of pro-inflammatory cytokines including interleukin 6 (IL-6), interleukin 1β (IL-1β) and tumour necrosis factor α (TNFα) which has been described as cytokine storm (CS) [1]. CS is a life-threatening condition that can lead to extensive tissue damage. In patients with severe COVID-19, CS is associated with lung injury, multi organ failure and poor prognosis [2–4]. Thus, early recognition and management of CS by targeting cytokines and modulating immune response could improve the outcome in these patients [1, 5]. While effective treatment options for COVID-19 remain limited, this clinical approach is widely used in many institutions.

IL-6 is one of the key mediators of the immune system and recently has been reported as a critical cytokine in COVID-19 associated CS [6]. While several studies have shown that elevated concentrations of IL-8 and TNFα correlate with severity of COVID-19 [4, 7, 8], IL-6 dysregulations appear to have the most profound effect in CS, with high concentrations associated with respiratory failure, poor prognosis and mortality [4, 6, 9, 10].

C-reactive protein (CRP) is an inflammatory marker that is raised in majority of patients with COVID-19 [11]. However, CRP does not necessarily indicate bacterial infection, with the highest concentrations commonly seen in more severe cases [11]. The major limitation of CRP is its low specificity in differentiating bacterial infection from autoimmune diseases and some haematological malignancies [12].

Procalcitonin (PCT) is an acute phase reactant peptide that increases significantly in the presence of bacterial infections while its concentrations are usually not elevated in viral infections [13]. Thus, PCT has been suggested as a useful biomarker to differentiate bacterial and viral infections with concentration ≥0.5 μg/L suggestive of a possible bacterial infection [14]. Furthermore, decrease in PCT could also indicate a good response to antibiotic therapy [15]. Nonetheless, similar to CRP, elevated PCT is not 100% specific for bacterial infections and can be seen in other systemic inflammatory responses [12]. For example, following major surgery, trauma, burns, invasive fungal infections and prolonged septic shock [14].

Several studies have shown that patients with more severe COVID-19 have higher PCT concentrations. Yet, it is not clear whether this is related to bacterial superinfection, severity of viral infection or combination of both [16]. Low specificity of PCT and its relatively long half-life have limited its use [12]. Using cytokines in combination with PCT may help to identify bacterial superinfection in COVID-19 [12]. Holub *et al.* reported that PCT, IL-6 and TNFα concentrations were significantly higher in bacterial infection compared to viral infection [12].

The same study showed that raised IL-6 and TNFα dropped within three days of antibiotic therapy to concentrations seen in control group, which included healthy participants [12].

In 2020, we introduced a cytokine panel (IL-1β, IL-6, IL-8 and TNFα) in our laboratory at King's College Hospital, UK, in response to the pandemic to help in the assessment of COIVD-19 patients. Each component in the panel was selected based on published literature on SARS-CoV-2 infection [17, 18]. The panel was validated within our laboratory as per our standard operating procedure with meets the medical laboratory standards (ISO 15189) of the National Accreditation Body for the United Kingdom (UKAS). While IL-6 assay is more widely available in the UK, the above cytokine panel is unique to King's College Hospital. The panel was offered for patients with suspected COVID-19 as part of their laboratory work-up.

The aim of our study was to compare PCT and CRP with the cytokine panel between ICU and non-ICU patients admitted to a London teaching hospital during the COVID-19 pandemic. Cytokine data was also compared with healthy volunteers.

## Methods

We retrospectively reviewed the laboratory information management system (LIMS–Clinisys) to identify patients admitted to King's College Hospital NHS Foundation Trust, London, a 950 bedded tertiary hospital, between March 2020 and January 2021, with positive SARS-CoV-2 RT-PCR and having had a cytokine panel and PCT requested as part of their clinical care. Lambda and Gamma variants of SARS-CoV-2 were more prevalent in London during the initial study period followed by Alpha variant in the latter months. The study was approved by the King's College Hospital Clinical Audit Committee (Ref ENDOC01) and the need for informed consent was waived. Medical records were accessed between June and December 2021. All data was anonymised prior to analysis and confidentially was maintained during analysis and storage.

102 ICU patients from COVID-19 ICU, and 62 age-matched non-ICU patients were randomly selected during the study period and their electronic medical records were reviewed. The primary reason for hospital admission of the patients in this study was suspected COVID-19 infection based on clinical/radiological evidence, which was subsequently confirmed with positive SARS-CoV-2 RT-PCR. 75 patients were excluded from the study as they were either enrolled into COVID-19 clinical trials or were on Remdesivir or IL-6 antagonist as part of COVID-19 treatment which could modify cytokine concentrations. Patients who did not have same day PCT with cytokine panel were also excluded to minimise variations in patients' individual infective state. A total of 46 ICU patients and 30 non-ICU patients were included in final analysis along with 24 healthy volunteers (Fig 1).

Healthy volunteers were recruited from our laboratory. They had no known medical conditions, had normal body mass index (BMI), were not on any medications and were asymptomatic at the time of blood sampling. All volunteers provided written consent as part of our evaluation process of the cytokine panel. This recruitment took place in April 2020, when diagnostic testing for asymptomatic individuals was not implemented in the UK. However, neither the volunteers nor their households had the common symptoms of COIVD-19 (high temperature, new or persistent cough, shortness of breath, loss of taste or loss of smell). Furthermore, their CRP was 1.00 (1.00–1.75) mg/L median (IQR) which suggested that they did not have active inflammation.

The local policy for ICU admission was based on the 40% (10L) or more oxygen requirement of the patient or rapid increase in oxygen requirements. In this study, all ICU patients receiving mechanical ventilation. Based on our ICU clinical guidelines, patients were divided into two groups; those with elevated PCT $\geq$ 0.5 and those with normal PCT < 0.5 μg/L. All patients (ICU + non-ICU) were on Empiric Antibiotic therapy. Bacterial infection was

> 164 patients selected (102 randomly selected ICU + 62 age-matched non-ICU)

> 75 excluded (55 ICU and 20 non-ICU) as either, enrolled in clinical trials
>
> or on Remdesivir or IL-6 antagonist

> 89 patients remaining

> 13 excluded due to not having same day PCT and cytokine data

> 76 patients include in the final analysis (46 ICU and 30 non-ICU) along with
>
> 24 healthy volunteers

**Fig 1. Flow chart of patient selection based on inclusion and exclusion criteria.**

identified base on positive cultures within 48 hours from the request for PCT and cytokine panel. In our cohort, bacterial infections occurred ≥48 hours following admission to hospital for COVID-19 thus, these infections are defined as superinfections. Among all patients admitted with COVID-19, 75% (58% ICU and 96% non-ICU) were on glucocorticoid as part of COVID-19 treatment.

## Sample collection

For cytokine panel, samples were collected in a serum separator tube (BD Vacutainer SST). All samples were kept at room temperature for 30 minutes prior to centrifugation for 15 minutes at 1000 x g. Serum aliquots were stored at -80°C and analysed within two days. IL-1β, IL-6, IL-8 and TNFα concentrations were quantified using the Simple Plex$^{TM}$ Ella (Ella$^{TM}$) (Protein-Simple, Bio-Techne, Oxford, UK), an automated immunoassay platform that allows the rapid quantitation of these four analytes from a single disposable microfluidic cartridge. PCT samples were collected in SST or lithium heparin plasma (BD Vacutainer) and analysed by Roche Cobas (Burgess Hill, UK).

## Cytokine panel performance characteristics

**Quality Control (QC) material.** Bio-Techne cytokine quality controls (QC) include a high and a low level, which were analysed with every assay. Each QC has predetermined values that are reagent lot specific. Performance was further assured through the linear standard curve with a percent coefficient of variance (%CV) of < 10% and a recovery of 80–120%.

**Limits of Quantification (LoQ).** The lower and upper limits of quantification in serum diluted 1:2 were; IL-1β (0.32–3060 pg/ml), IL-6 (0.56–5304 pg/ml), IL-8 (0.38–3608 pg/ml) and TNFα (0.6–2320 pg/ml).

**Endogenous concentrations.** The serum endogenous cytokine concentrations calculated from the 24 healthy volunteers were; IL-1β (0.00–0.66 pg/ml), IL-6 (0.00–3.26 pg/ml), IL-8 (2.20–21.87 pg/ml) and TNFα (6.10–13.58 pg/ml).

**Precision.** The %CV of inter-assay precision for all four cytokines ranged between 3–5% for low QC and 4–9% for high QC. The % CV for intra-assay precision ranged between 3–5% for low QC and 4–9% for high QC.

**Correlation.** Ella™ IL-6 correlation with IMMULITE® 2000 IL-6 (Siemens Healthineers, Frimley, UK) and Evidence Investigator™ (Randox, Crumlin, UK). Cytokine and Growth Factors High-Sensitivity Array showed comparable data with $R^2$ values of 0.98 and 0.98 respectively.

**Freeze/Thaw cycle.** The effects of up-to two freeze thaw cycles on cytokine concentrations were evaluated for IL-6, IL-8 and TNFα. The results from one and two freeze thaw cycles were comparable with $R^2$ values of 0.99. All test samples included in the final analysis underwent only one freeze thaw cycle.

## Statistical analysis

Analyse IT (Microsoft, version 5.2) was used for statistical analysis. Distribution of the data were assessed using Shapiro- Wilk test and were analysed by non-parametric Mann Whitney U or Kruskul Wallis test as appropriate. Chi-square was performed for binary data. Spearman's Rank Correlation Coefficient was used for correlations between analysts. A p value < 0.05 was considered a statistically significant result. Data is reported as mean (SD) or median (interquartile range (IQR)).

## Results

The final analysis of the clinical cohort included, 24 healthy volunteers aged 33 (7) years, 46 ICU patients and 30 non-ICU age-matched (55 (9) years) patients. 56% of ICU patients, 63% of non-ICU patients and 38% of healthy volunteers were males.

## Cytokine panel in patients with COVID-19 vs. healthy volunteers

IL-1β, IL-6, IL-8, TNFα and CRP in the hospitalised patients with COVID-19 (n = 76) was higher (all p<0.0001) compared to the healthy volunteers. The higher values vs. the healthy volunteers were noted in COVID-19 patients irrespective of ICU or non-ICU admission (Table 1).

## Cytokine panel in ICU patients and non-ICU patients with COVID-19

The length of stay for ICU patients was 35 (23–59) days. The total duration of hospital admission for ICU patients was longer 60 (32–87) days compared to non-ICU patients 10 (7–15) days, p<0.00001.

Overweight/obesity (BMI ≥ 25 kg/m$^2$), followed by hypertension and diabetes were the most common co-morbidities in our COVID-19 cohort. The BMI and prevalence of hypertension, diabetes, lung disease, cardiovascular disease and chronic kidney disease were similar between ICU and non-ICU patients (Table 2).

The comparison of IL-1β, IL-6, IL-8 and TNFα, PCT and CRP concentrations in both groups are summarised in Table 3. IL-6, IL-8, TNFα and PCT were higher in ICU patients vs. non-ICU patients (all, p<0.05). However, there was no difference in CRP between the two groups. In non-ICU group, only four patients (13%) had high PCT (≥ 0.5 μg/L) compared to 26 (57%) in ICU patients.

**Table 1. Cytokine panel and CRP between ICU, non-ICU patients and healthy volunteers.**

| Analyte | ICU patients | Non-ICU patients | Healthy Volunteers | P value |
|---|---|---|---|---|
| | (n = 46) | (n = 30) | (n = 24) | |
| **IL-1β (ng/L)** | 0.35 (0.32–0.66) | 0.32 (0.32–0.32) | 0.11 (0.05–0.21) | <0.0001 |
| **IL-6 (ng/L)** | 36.50 (16.07–127.50) | 17.45 (7.90–51.60) | 0.99 (0.62–1.42) | <0.0001 |
| **IL-8 (ng/L)** | 72.20 (49.47–115.50) | 40.60 (29.00–60.50) | 10.55 (7.58–15.67) | <0.0001 |
| **TNFα (ng/L)** | 25.00 (18.08–35.75) | 16.35 (14.10–19.80) | 9.83 (8.40–11.27) | <0.0001 |
| **CRP (mg/L)** | 92.5 (54.9–137.3) | 105.0 (68.2–146.2) | 1.00 (1.00–1.7) | <0.0001 |

Data is presented as median (IQR).

Abbreviations: IL-1β, interleukin-1β; IL-6, interleukin-6; IL-8, interleukin-8; TNFα, tumour necrosis factor alpha; CRP, C-reactive protein.

Since fewer patients (58%) in ICU were receiving glucocorticoid compared to non-ICU (96%) patients, we compared cytokine concentrations, PCT and CRP between ICU patients on glucocorticoids (n = 27) vs. those not on glucocorticoids (n = 19) and there were no differences (all, p>0.05). We also compared ICU patients on glucocorticoids (n = 27) with non-ICU patients on glucocorticoids (n = 29). The IL-6, IL-8, TNFα and PCT were higher in ICU patients (all, p<0.05) with no difference in CRP or IL-1β between the two groups (both, p>0.05).

## Cytokine panel in ICU patients with COVID-19 (high PCT vs. normal PCT)

Among ICU patients, 26 (56%) had a PCT $\geq$ 0.5 μg/L. Cytokine and CRP in patients with high PCT ($\geq$ 0.5 μg/L) and normal PCT (< 0.5 μg/L) are summarised in Table 4. TNFα concentration was higher in high PCT group vs. normal PCT group, p<0.01. Patients with high PCT had longer ICU stay 47 (27–64) vs. 25 (20–38) days, p<0.05.

At the time of sample collection for the cytokine panel, all patients were receiving broad-spectrum antibiotics and in total nine (19%) were on adjunctive anti-fungal treatment (25% in normal PCT group and 15% in high PCT group). Forty eight percent of patients in ICU had positive culture for bacterial growth within 48 hours from PCT and cytokine requests compared to 10% in the non-ICU group. Out of all positive cultures, 84% were respiratory samples (sputum, bronchial washing and bronchoalveolar lavage), 8% were blood samples and 8% were urine samples. The most common pathogens identified from cultures were Gram

**Table 2. Common co-morbidities in ICU vs. non-ICU patients with COVID-19.**

| Comorbidities | ICU patients | Non-ICU patients | P value |
|---|---|---|---|
| | (n = 46) | (n = 30) | |
| **BMI $\geq$ 25 kg/m$^2$** | 31 (67%) | 17 (57%) | 0.34 |
| **Hypertension** | 17 (37%) | 14 (47%) | 0.40 |
| **Diabetes** | 11 (24%) | 10 (33%) | 0.37 |
| **Lung disease[*]** | 9 (20%) | 5 (17%) | 0.75 |
| **CVD** | 5 (11%) | 5 (17%) | 0.46 |
| **CKD** | 7 (15%) | 3 (10%) | 0.51 |

Abbreviations: BMI, body mass index; CVD, cardiovascular disease; CKD, chronic kidney disease.

[*]Patients with background of lung disease had one of the following: Chronic obstructive pulmonary disease (COPD), Obstructive sleep apnoea (OSA), interstitial lung disease (ILD) and Asthma (moderate and severe)

**Table 3. Cytokines, PCT and CRP in ICU vs. non-ICU patients with COVID-19.**

| Analyte | ICU patients | Non-ICU patients | P value |
|---|---|---|---|
| | (n = 46) | (n = 30) | |
| IL-1β (ng/L) | 0.35 (0.32–0.66) | 0.32 (0.32–0.32) | 0.07 |
| IL-6 (ng/L) | 36.50 (16.07–127.50) | 17.45 (7.90–51.60) | **0.03** |
| IL-8 (ng/L) | 72.20 (49.47–115.50) | 40.60 (29.00–60.50) | **<0.001** |
| TNFα (ng/L) | 25.00 (18.08–35.75) | 16.35 (14.10–19.80) | **<0.0001** |
| PCT (μg/L) | 0.65 (0.30–1.98) | 0.18 (0.12–0.29) | **<0.0001** |
| CRP (mg/L) | 92.5 (54.9–137.3) | 105.0 (68.2–146.2) | 0.77 |

Data is presented as median (IQR).

Abbreviations: PCT, procalcitonin; IL-1β, interleukin-1β; IL-6, interleukin-6; IL-8, interleukin-8; TNFα, tumour necrosis factor alpha; CRP, C-reactive protein.

negative bacteria including *Klebsiella pneumoniae*, *Pseudomonas aeruginosa*, *Serratia marcescens*, *Enterobacter cloacae and E.coli*. The only two gram-positive bacteria that was isolated were *Staphylococcus aureus* (sputum) and *Enterococcus faecium* (urine). There was no difference in rate of positive cultures between the high PCT and normal PCT ICU patients (p = 0.73).

There was no difference in age (p = 0.58) or BMI (p = 0.76) between female and male patients admitted to ICU. However, males had higher concentrations of PCT (p<0.00001) and TNFα (p<0.001) and longer ICU stay (p<0.01) compared to the females.

## Correlation of cytokine panel, PCT and CRP

In ICU patients, PCT concentrations correlated with TNFα (r = 0.79, p<0.0001), IL-1β (r = 0.36, p = 0.009) and IL-8 (r = 0.43, p = 0.02) while CRP correlated with both IL-6 (r = 0.66, p<0.001) and IL-1β (r = 0.42, p = 0.02). In non-ICU patients, PCT correlated only with TNFα (r = 0.37, p = 0.04). No correlations were observed between CRP and the cytokines.

## Discussion

We have demonstrated that using a cytokine panel (pro-inflammatory cytokines; IL-1β, IL-6, IL-8 and TNFα) in combination with PCT could be useful in the assessment SARS-CoV-2 infection severity. IL-6, IL-8, TNFα and PCT were significantly higher in ICU patients compared to non-ICU patients. Furthermore, ICU patients with elevated PCT had higher TNFα and longer ICU stay compared to normal PCT group. In our ICU cohort, males had higher TNFα, PCT and longer ICU stay with COVID-19 compared to females.

Cytokines are essential part of the host immune response against various pathogens. IL-1, IL-6 and TNFα are three main pro-inflammatory cytokines that are produced by endothelial and epithelial cells, macrophages and mast cells during innate immune response against viral infection [1]. Furthermore, IL-1, IL-6 and TNFα, have been reported as the pathogenic factors produced by macrophages after T lymphocytes bearing T-cell receptors recognise SARS-CoV-2 bound to the surface of cells [19]. SARS-CoV-2 enter the bronchial epithelium of the upper and lower airway and provoke local inflammatory cascades that involve neutrophil recruitment, T lymphocyte trafficking and activation of resident monocytes. The virus activates the humoral immune response to cause increased secretion of pro-inflammatory cytokines [20]. These cytokines maybe responsible for tissue destruction in various organs in patients with COVID-19 [19].

**Table 4. Cytokines and CRP in ICU patients with high and normal PCT.**

| Analyte | PCT ≥ 0.5 µg/L | PCT < 0.5 µg/L | P value |
|---|---|---|---|
| | (n = 26) | (n = 20) | |
| IL-1β (ng/L) | 0.45 (0.32–1.03) | 0.32 (0.33–0.40) | 0.12 |
| IL-6 (ng/L) | 44.95 (25.85–119.75) | 25.30 (14.97–130.50) | 0.41 |
| IL-8 (ng/L) | 72.20 (51.05–116.57) | 69.00 (47.70–112.50) | 0.92 |
| TNFα(ng/L) | 33.55 (24.92–38.92) | 20.25 (13.97–23.35) | **<0.01** |
| CRP (mg/L) | 108.6 (60.0–223.7) | 81.8 (54.9–106.5) | 0.11 |

Data is presented as median (IQR).

Abbreviations: PCT, procalcitonin; IL-1β, interleukin-1β; IL-6, interleukin-6; IL-8, interleukin-8; TNFα, tumour necrosis factor alpha; CRP, C-reactive protein

SARS-CoV-2 infection in some patients can trigger an acute exaggerated immune response with sudden increase in pro-inflammatory cytokines, which is known as CS [1, 21, 22]. In our study, selected inflammatory cytokines were analysed as potential biomarkers to help identify virally driven inflammation and bacterial superinfection.

Differentiating between inflammatory response driven by the virus and secondary bacterial infection could be challenging in managing patients with COVID-19. High CRP is seen in majority of patients with COVID-19 with higher concentrations associated with severity of COVID-19 and it does not necessarily help identify bacterial superinfection [11]. A meta-analysis of four studies showed PCT concentrations were associated with more severe COVID-19 infection [16]. However, in patients with non-complicated COVID-19, PCT could remain within normal reference limits [16]. The authors suggested that elevated PCT in patients developing severe COVID-19 could possibly reflect bacterial superinfection [16]. In accordance with this meta-analysis, our study showed that ICU patients had higher PCT compared to non-ICU patients. Furthermore, ICU patients with PCT >0.5 µg/L had longer ICU stay vs. the low PCT group, whilst there was no difference in CRP between the two groups.

PCT has a relatively long half-life, which could limit its use [12]. Using cytokines in addition to PCT may help in identifying bacterial infections in COVID-19. We also reviewed the correlation of cytokines with CRP and PCT in order to assess whether they will equally assist in identifying bacterial infections. Holub *et al*, analysed several inflammatory cytokines along with CRP and PCT in 21 patients with community acquired pneumonia and 26 patients with viral infections [12]. Their results showed that IL-6 and TNFα were higher in bacterial infections compared to viral infections and elevated cytokine concentrations dropped within three days of antibiotic therapy [12]. The same study also showed positive correlation between TNFα and PCT which was in keeping with our data in the COVID-19 cohort. In addition, in our cohort both IL-1β and IL-8 correlated positively with PCT. Nonetheless, these correlations were weak and larger studies are required to better assess them.

Gong *et al*, analysed inflammation markers in 100 patients suffering from mild, severe and critical COVID-19 infection and reported that IL-8 concentrations were associated with COVID-19 severity [7]. In our study, high PCT group had higher TNFα and longer ICU stay and there was a positive correlation between PCT and TNFα. In clinical setting, this positive correlation may strengthen the predictability of the disease severity. One possible explanation for the high PCT group requiring longer ICU stay could be bacterial superinfection. However, no difference was found between the rates of confirmed bacterial infections based on positive cultures between high vs. normal PCT groups. This could have been due to treating patients with antibiotics prior to requesting samples for cultures. Our results were similar to the study by Hu *et al*. in which PCT was associated with disease severity however; the percentage of

bacterial infections in severe and critical COVID-19 were lower than the percentage of elevated PCT [22]. Thus, larger studies are needed to investigate the underlying mechanisms driving PCT concentrations in SARS-CoV-2 infection.

CRP production and release from liver is stimulated by IL-6. Additionally, IL-6 and CRP have been associated with COVID-19 severity, with IL-6 being the most frequently reported cytokine elevated in COVID-19 associated CS [21, 23]. Furthermore, higher IL-6 has been shown to be associated with severity and higher mortality rate in COVID-19 [4, 6, 24, 25]. The Randomised Evaluation of COVID-19 Therapy (RECOVERY trial) reported that treatment with Tocilizumab (IL-6 receptor antagonist), improves outcome in patients hospitalised with severe COVID-19 infection [26]. This effect was additional to benefits previously reported for Dexamethasone treatment in patients hospitalized with COVID-19 [24]. The National Institute for Health and Care Excellence (NICE) guidelines has recommended Tocilizumab treatment in patients with COVID-19 receiving Oxygen treatment with a CRP of $\geq$ 75 mg/L provided there is no evidence of bacterial or other viral infections that could be worsened by immunosuppression [27]. A prospective cohort study of 89 patients showed that high concentrations of IL-6 followed by CRP could predict respiratory failure and the need for mechanical ventilation in patients admitted with COVID-19 with significant time difference between CRP and IL-6 in favour of IL-6 [28]. In our study however, despite significant increase in IL-6 in ICU patients vs. non-ICU patients, there was no difference in CRP between the two groups.

IL-1 is another important cytokine family in CS. One of the actions of IL-1β, which belongs to the IL-1 family, is upregulation of IL-6. Thus, the IL-1/IL-6/CRP axis plays a key role in the inflammation process [29]. In our COVID-19 cohort, CRP correlated positively with both IL-6 and IL-1β but there was no significant correlation between PCT and IL-6. It could be that the severity of COVID-19 in these patients with high IL-6 was due to virus driven inflammation rather than bacterial superinfection.

Our study has several limitations. It was a retrospective study design involving a small cohort. The healthy volunteer group was younger than the patient groups and their COVID-19 status was not confirmed by RT-PCR. We did not match for common co-morbidities between ICU and non-ICU patients however, the overall rate of co-morbidities between the two groups were similar.

## Conclusions

We have reported the application of a cytokine panel (serum, IL-1β, IL-6, IL-8 and TNFα) using the automated Ella™ platform and its potential clinical utility in the assessment of patients with COVID-19. In our cohort, ICU patients had higher IL-6, IL-8, TNFα and PCT compared to non-ICU patients. Despite no difference in CRP or rate of confirmed bacterial superinfection, ICU patients with high PCT had higher TNFα and longer ICU admission with COVID-19. Further prospective studies are needed in larger well-defined cohorts to establish the routine use of this cytokine panel in an ICU setting and to elucidate the mechanisms by which PCT is increased in severe COVID-19.

## Supporting information

**S1 File. Dataset.**
(XLSX)

## Author Contributions

**Conceptualization:** Tina Mazaheri, Royce P. Vincent.

**Data curation:** Tina Mazaheri, Allison Manning.

**Formal analysis:** Tina Mazaheri, Wiaam Al-Hasani, Jessica Kearney.

**Methodology:** Royce P. Vincent.

**Supervision:** Royce P. Vincent.

**Validation:** James Luxton, Tracey Mare.

**Writing – original draft:** Tina Mazaheri.

**Writing – review & editing:** Ruvini Ranasinghe, Georgios K. Dimitriadis, Royce P. Vincent.

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
