## [Decision Letter · Decision Letter 0]

27 Oct 2021

PONE-D-21-26044A cytokine panel and procalcitonin in COVID-19, a comparison between intensive care and non-intensive care patientsPLOS ONE

Dear Dr. Mazaheri,

Thank you for submitting your manuscript to PLOS ONE. After careful consideration, we feel that it has merit but does not fully meet PLOS ONE’s publication criteria as it currently stands. Therefore, we invite you to submit a revised version of the manuscript that addresses the points raised during the review process.

ACADEMIC EDITOR: Please review comments made by reviewers and provide a point by point response in your revised manuscript.

We look forward to receiving your revised manuscript.

Kind regards,

Muhammad Adrish, MD, MBA, FCCP, FCCM

Academic Editor

PLOS ONE

Journal Requirements:

2. In the ethics statement in the manuscript and in the online submission form, please provide additional information about the patient records used in your retrospective study, including: a) whether all data were fully anonymized before you accessed them; b) the date range (month and year) during which patients' medical records were accessed; c) the date range (month and year) during which patients whose medical records were selected for this study sought treatment. If the ethics committee waived the need for informed consent, or patients provided informed written consent to have data from their medical records used in research, please include this information.

Reviewers' comments:

Reviewer's Responses to Questions

**Comments to the Author**

1. Is the manuscript technically sound, and do the data support the conclusions?

Reviewer #1: Partly

Reviewer #2: Partly

2. Has the statistical analysis been performed appropriately and rigorously? 

Reviewer #1: I Don't Know

Reviewer #2: I Don't Know

3. Have the authors made all data underlying the findings in their manuscript fully available?

Reviewer #1: Yes

Reviewer #2: Yes

4. Is the manuscript presented in an intelligible fashion and written in standard English?

Reviewer #1: Yes

Reviewer #2: No

5. Review Comments to the Author

Reviewer #1: Some suggestions:

Abstract:

- in general, the "intensive care unit" is referred to the "ICU". Many health care system will also indicate whether the patient will be managed in the different specific ICUs, like medical ICU (MICU) or surgical ICU (SICU), but typically we refer to this level of care as an ICU. In the UK, where this study was done, ITU refers to the "intensive therapy unit" which from what I am reading is a direct correlate to the ICU as described in areas of the world like the United States. Since this publication is likely going to reach a global audience, I would suggest using ICU throughout the manuscript but if ITU is kept then the abbreviation should be placed in the title and "intensive therapy unit" should be used throughout.

Methods:

- Line 122: please indicate how many patient beds are available at the hospital and the exact name of the Hospital where the study was conducted

- In the methods section, you should include what were the general requirements for those admitted to the ICU at your hospital. For example, what level of oxygen supplementation was the minimum requirement to be admitted? Were these all patients placed on mechanical ventilation? I would include that in you methods and would even consider to include in the analysis.

- a small flow diagram of those inlcuded/excluded and reasons would be useful for the readers

- From the 63 age-matched non-ITU patients, how did you make that match? Did you match other chronic conditions? There could be a sampling bias at hand but this should be discussed.

- Do we know who was on glucocorticoids and who was not?

- Do we know which SARS-CoV-2 variant was circulating in this regions during the study period? I would include that.

Results:

- I think it is very important to discuss the clinical factors of those with and without COVID-19 as it relates to the cytokine and CRP testing; it is great to compare to healthy volunteers as a control but were they tested for COVID-19? Could they have been asymptomatically infected? Please explain the recruitment of the healthy volunteers.

- Common comorbidities to consider in the analysis would be: Diabetes Mellitus, Hypertension, Obesity, Chronic Kidney Disease, Chronic Obstructive Pulmonary Disease, and others processes.

Discussion:

- I would add some discussion on IL-6 and CRP in the clinical context of COVID-19 and initiation of certain immune-modulating medications, like tocilizumab, or barcitinib, among others, who are found to have high initial CRP and IL-6. In the U.S., guidelines are now suggesting use in certain populations who have CRP greater than or equal to 75 mg/L and rapidly increasing oxygen requirements. I would also cite a large trial like the RECOVRY trial which was conducted in the UK. NHS also recommends IL-6 inhibitors in certain populations. A discussion on the results as it pertains to clinical use is suggested.

Limitations needs attention. Do we know who was glucocorticoids (I would not use steroids). Small sample size is a big limitation and needs to be acknowledged. What does "some patients were requiring organ support" mean? This is why it is important to assess how patients were matched, what level of ICU care was needed and comorbidities of the cohort.

Reviewer #2: This article discussed relationships between cytokines, procalcitonin, and C-reactive protein among COVID-19 patients in London during certain months of the pandemic. The authors compared cytokines, PCT, and CRP between COVID-19 patients and healthy volunteers and also between ITU patients and non-ITU patients. The authors suggest previous research with cytokines to help determine any difference between viral infection and bacterial infection. The authors also admit that further research is needed to assess the relationship between findings from a cytokine panel and COVID-19.

Investigation of the inflammatory response among COVID-19 patients, such as those in an ITU setting, is important for clinical practice. While the intentions of this study are valid, it is difficult to follow the main aim and resulting comparisons included in the analysis. The Methods need to be clearer, especially with defining the study population and inclusion terms, and the overall organization of the paper needs improvement.

Major Comments:

In the Introduction, include a description or paragraph about the “recently validated cytokine panel” as stated in lines 115-116. Give more background information on this panel, as this cytokine panel is one of main topics of this paper. How was this particular panel validated? How was this panel used previously? Briefly, how does this panel work?

The aim of the paper states that there were assessments of concurrent bacterial infections with COVID-19; however, there is a lack of descriptions about concurrent bacterial infections. In the Methods section, include how bacterial infections were assessed. Which bacterial testing was used? Which bacteria could be detected on these tests? When were patients included in this study tested for concurrent bacterial infection? In the Discussion, it is unclear how the findings are directly related to the main aim of assessing bacterial co-infection.

The Methods and Results sections include comparisons with 24 healthy volunteers; however, it is unclear how the volunteers were chosen for this study. How were these volunteers chosen? Were these volunteers matched at all (the non-ITU patients were age-matched)? When were these volunteers sampled? Were the volunteers sampled during the same timeframe as the other patients? Describe the criteria needed to be considered “healthy.” Also, the Introduction sets up the study to compare ITU and non-ITU patients, so some explanation about the COVID-19 patients versus “healthy” volunteers is needed earlier in the paper. Were comparisons made between ITU patients, non-ITU patients, and “healthy” volunteers? A three-way comparison with “healthy” volunteers as the referent may be more elucidating.

Minor Comments:

Line 87: Make a new paragraph for CRP descriptions.

Lines 111-113: Clarify which study is referenced here and describe the control group used in that study. Introduce the study more clearly earlier in this paragraph.

Lines 117-118: Give some context about COVID-19 incidence in London during the study period and how the cytokine panel was used at this particular hospital. Give the months/year of the “peak” pandemic time.

Line 122: State the specific months/year that included the patients in this study.

Line 123: When were patients tested for SARS-CoV-2? Was there a specific timeframe from exposure that these patients were tested? Was the testing implemented for diagnostic purposes or as a general hospital screening test during that time period?

Line 127: Specify how many of the 75 excluded patients were ITU or non-ITU.

Line 130: Explain why patients needed to have a same-day PCT result as the cytokine panel to be included in the study. Give the number of patients excluded based on not having same-day tests—was it 14 patients (make sure that the total numbers reflect the 76 included patients)?

Consider making a flow chart delineating the inclusion/exclusion factors for included patients.

Lines 142-144: Is this the test that was mentioned within the introduction?

Line 152: Spell out CV when it is first mentioned.

Table 1: The row for PCT is missing. Add PCT results.

Line 211: Indicate that the finding was statistically significant.

Line 247: Interleukin is misspelled.

Within the Discussion, explain why correlations were used for PCT and CRP among the ITU and non-ITU patients. Even though the correlations are statistically significant, most correlations seemed weak. How do these findings translate to clinical practice?

Line 269: How was “severity” defined? How can these higher cytokines be shown with certain clinical signs in patients?

Line 289: Add an in-text citation.

Lines 291-292: How were concurrent bacterial infections confirmed in this reference study and also in the present study?

Line 304: Add an in-text citation.

Lines 311-314: Please re-phrase this sentence, as it is difficult to follow.

Line 315: Are the “two groups” mentioned here the high and low PCT groups?

Line 328: Please specify the “two values.”

Line 355: State the number of patients (46) instead of “good number.”

There are many typographical errors, such as missing hyphens and inconsistent hyphenation, missing commas, missed spacing between words, missing articles before nouns, and missing or incorrect punctuation.

6. PLOS authors have the option to publish the peer review history of their article (what does this mean?). If published, this will include your full peer review and any attached files.

Reviewer #1: No

Reviewer #2: No

---

## [Author Response · Author response to Decision Letter 0]

10 Jan 2022

Reviewer #1: Some suggestions:

Abstract:

- in general, the "intensive care unit" is referred to the "ICU". Many health care system will also indicate whether the patient will be managed in the different specific ICUs, like medical ICU (MICU) or surgical ICU (SICU), but typically we refer to this level of care as an ICU. In the UK, where this study was done, ITU refers to the "intensive therapy unit" which from what I am reading is a direct correlate to the ICU as described in areas of the world like the United States. Since this publication is likely going to reach a global audience, I would suggest using ICU throughout the manuscript but if ITU is kept then the abbreviation should be placed in the title and "intensive therapy unit" should be used throughout.

Response: Thank you for the suggestion. We have revised the manuscript to the preferred terminology for intensive care, ICU throughout the article.

Methods:

- Line 122: please indicate how many patient beds are available at the hospital and the exact name of the Hospital where the study was conducted. 

Response: We have now included the name and bed capacity of our hospital – lines 121 -122.

- In the methods section, you should include what were the general requirements for those admitted to the ICU at your hospital. For example, what level of oxygen supplementation was the minimum requirement to be admitted? 

Response: The admission to ICU was considered when the oxygen requirement of the patient was 40% (10L) or more or those who show a rapid increase in oxygen requirement – lines 155-156.

Were these all patients placed on mechanical ventilation? I would include that in your methods and would even consider to include in the analysis.

Response: All ICU patients were receiving mechanical ventilation. Included in method section– lines 156-157.

- a small flow diagram of those included/excluded and reasons would be useful for the readers

Response: We have now included the flow diagram as suggested – figure 1.

- From the 63 age-matched non-ITU patients, how did you make that match? Did you match other chronic conditions? There could be a sampling bias at hand but this should be discussed.

Response: We only matched the groups for age. Chronic conditions were not matched between ICU and non-ICU patients. However, the common chronic conditions like Diabetes, Hypertension, lung disease, chronic kidney disease and cardiovascular disease between two groups were similar and have now added in results section Table 2, Page 11.

- Do we know who was on glucocorticoids and who was not?

Response: This has now been included in lines 166-167. A new Paragraph has now added in Results section because fewer patients in ICU were on glucocorticoids compared to non-ICU patients–lines 281-287. 

- Do we know which SARS-CoV-2 variant was circulating in this regions during the study period? I would include that.

Response: This has now been included in lines 127-129.

Results:

- I think it is very important to discuss the clinical factors of those with and without COVID-19 as it relates to the cytokine and CRP testing; it is great to compare to healthy volunteers as a control but were they tested for COVID-19? Could they have been asymptomatically infected? Please explain the recruitment of the healthy volunteers.

Response: Details around the recruitment of the healthy volunteers has now been included – lines 149-156.

- Common comorbidities to consider in the analysis would be: Diabetes Mellitus, Hypertension, Obesity, Chronic Kidney Disease, Chronic Obstructive Pulmonary Disease, and others processes.

Response: This has now been included in results-line 247-251 and Table 2.

Discussion:

- I would add some discussion on IL-6 and CRP in the clinical context of COVID-19 and initiation of certain immune-modulating medications, like tocilizumab, or barcitinib, among others, who are found to have high initial CRP and IL-6. In the U.S., guidelines are now suggesting use in certain populations who have CRP greater than or equal to 75 mg/L and rapidly increasing oxygen requirements. I would also cite a large trial like the RECOVRY trial which was conducted in the UK. NHS also recommends IL-6 inhibitors in certain populations. A discussion on the results as it pertains to clinical use is suggested.

Response: We have now included the conclusions of the RECOVERY trial and NICE guidelines recommendations for Tocilizumab–lines 391-399.

Limitations needs attention. Do we know who was glucocorticoids (I would not use steroids).

Response: 75% of patients (ICU & non-ICU) were on glucocorticoids as part of the management of COVID-19 and associated complications. This has now been included in the manuscript –lines 166-167 and lines 281-287. 

Small sample size is a big limitation and needs to be acknowledged. 

Response: We have included the small sample size as a limitation of this study. The numbers however represent a high percentage of patients to have had cytokine results vs. other hospitals in the UK treating large number of COVID-19 patients. This is due to the limited availability of routine cytokine measurement in UK labs.

What does "some patients were requiring organ support" mean? This is why it is important to assess how patients were matched, what level of ICU care was needed and comorbidities of the cohort.

Response:

All ICU patients required mechanical ventilation, which is now included – lines 159-160.

Co-morbidities between ICU/non-ICU groups is now included in table 2 and lines 247-251.

\f

Reviewer #2: 

This article discussed relationships between cytokines, procalcitonin, and C-reactive protein among COVID-19 patients in London during certain months of the pandemic. The authors compared cytokines, PCT, and CRP between COVID-19 patients and healthy volunteers and also between ITU patients and non-ITU patients. The authors suggest previous research with cytokines to help determine any difference between viral infection and bacterial infection. The authors also admit that further research is needed to assess the relationship between findings from a cytokine panel and COVID-19.

Investigation of the inflammatory response among COVID-19 patients, such as those in an ITU setting, is important for clinical practice. While the intentions of this study are valid, it is difficult to follow the main aim and resulting comparisons included in the analysis. The Methods need to be clearer, especially with defining the study population and inclusion terms, and the overall organization of the paper needs improvement.

Response: We thank the reviewer for the above comments. We have now re-structured the manuscript to address this and hope it now provides better clarity.

Major Comments:

In the Introduction, include a description or paragraph about the “recently validated cytokine panel” as stated in lines 115-116. Give more background information on this panel, as this cytokine panel is one of main topics of this paper. 

How was this particular panel validated? 

How was this panel used previously? Briefly, how does this panel work?

Response: We have now expended on the cytokine panel to clarify when it was introduced and the process uses to validate its routine use – Lines 109-115. More details about validation mentioned in Method section under Cytokine Panel Performance Characteristics - Lines 182 -208.

The aim of the paper states that there were assessments of concurrent bacterial infections with COVID-19; however, there is a lack of descriptions about concurrent bacterial infections. In the Methods section, include how bacterial infections were assessed. 

Which bacterial testing was used? Which bacteria could be detected on these tests? When were patients included in this study tested for concurrent bacterial infection? 

Response: This is now included in method section (Lines 163-165) and in result section (lines 305-308).

In the Discussion, it is unclear how the findings are directly related to the main aim of assessing bacterial co-infection.

Response: we have now simplified the aim of this study (Lines 117-119) as our data is observational for bacterial co-infection. Described in the discussion–Lines 359-385.

The Methods and Results sections include comparisons with 24 healthy volunteers; however, it is unclear how the volunteers were chosen for this study. 

How were these volunteers chosen? 

Were these volunteers matched at all (the non-ITU patients were age-matched)? 

When were these volunteers sampled? Were the volunteers sampled during the same timeframe as the other patients? 

Describe the criteria needed to be considered “healthy.” 

Response: The recruitment of the healthy volunteers has now been included – lines 149-156. 

The volunteers were younger than the COVID-19 patients which is now included in result section (lines 221-222) and limitations (lines 414-415). 

All volunteers sampled in April 2020 for the validation of the cytokine panel within our laboratory (line 151). A proportion of patients’ samples were collected during this period. 

Also, the Introduction sets up the study to compare ITU and non-ITU patients, so some explanation about the COVID-19 patients versus “healthy” volunteers is needed earlier in the paper. 

Response: We have now referred to the healthy volunteers in the abstract (line 47 and 51) and under Introduction (line 119).

Were comparisons made between ITU patients, non-ITU patients, and “healthy” volunteers? A three-way comparison with “healthy” volunteers as the referent may be more elucidating.

Response: We have now compared three groups as suggested (lines 229-232) and results are summarised in Table 1 

Minor Comments:

Line 87: Make a new paragraph for CRP descriptions.

Response: CRP description is now in separated paragraph - Lines 84 – 88.

Lines 111-113: Clarify which study is referenced here and describe the control group used in that study. Introduce the study more clearly earlier in this paragraph.

Response: Reference given and control group is described - Lines 105-108.

Lines 117-118: Give some context about COVID-19 incidence in London during the study period and how the cytokine panel was used at this particular hospital. Give the months/year of the “peak” pandemic time.

Response: Included in the methods (Lines 125 -129), Introduction (line 109 to 115). ‘Peak’ pandemic time has been omitted from the Introduction as not relevant to our study data.

Line 122: State the specific months/year that included the patients in this study.

Response: March 2020 to January 2021, included in the methods (Lines 125-126)

Line 123: When were patients tested for SARS-CoV-2? Was there a specific timeframe from exposure that these patients were tested? Was the testing implemented for diagnostic purposes or as a general hospital screening test during that time period?

Response: The reason for admission of all patients in this study was suspected COVID-19 infection based on clinical/radiological evidence which was confirmed with positive (RT-PCR) SARS-CoV-2. Included in method section (Lines 137-139).

Line 127: Specify how many of the 75 excluded patients were ITU or non-ITU.

Response: 55 ICU and 20 non-ICU, now included in Figure 1.

Line 130: Explain why patients needed to have a same-day PCT result as the cytokine panel to be included in the study. Give the number of patients excluded based on not having same-day tests—was it 14 patients (make sure that the total numbers reflect the 76 included patients)?

Response: Included in line 143 in method and Fig 1.

Consider making a flow chart delineating the inclusion/exclusion factors for included patients.

Response: We have now included a figure to highlight this (fig 1, page 6).

Lines 142-144: Is this the test that was mentioned within the introduction?

Response: No in these references, they have used different methods for PCT. 

Line 152: Spell out CV when it is first mentioned.

Response: Amended as suggested–Line 185.

Table 1: The row for PCT is missing. Add PCT results.

Response: PCT is not included in Table 1 as PCT was not measured in healthy volunteers. PCT is however included In Table 3.

Line 211: Indicate that the finding was statistically significant.

Response: we have made this change (Line 263).

Line 247: Interleukin is misspelled.

Response: This has now been corrected (Line 300).

Within the Discussion, explain why correlations were used for PCT and CRP among the ITU and non-ITU patients. Even though the correlations are statistically significant, most correlations seemed weak. How do these findings translate to clinical practice?

Response: We have suggested that correlation of cytokines with CRP and PCT may help to assist in identifying bacterial infections. Included under in discussion, lines 361 -363, 370-371, 375-376). We agree with the reviewer that the correlations are weak and larger studies are required to better assess this.

Line 269: How was “severity” defined? How can these higher cytokines be shown with certain clinical signs in patients?

Response: In general ICU patients with elevated PCT had higher TNFα and longer ICU length of stay. (Severity has been replaced by ICU length of stay- lines 330-333)

Line 289: Add an in-text citation.

Response: This has now been included–line 351 and 352.

Lines 291-292: How were concurrent bacterial infections confirmed in this reference study and also in the present study?

Response: In our study, concurrent bacterial infection has changed to concurrent and/or super added bacterial infection. Concurrent and/or super added bacterial infection were identified based on positive blood or respiratory cultures within 48 hours from cytokine panel request time. This is included under Methods - lines 163-175

In reference study, authors suggested significant increase in PCT could possibly reflect bacterial coinfection in those developing severe COVID-19 thus contributing to complicate its clinical picture, this is now included in lines 352-353)

Line 304: Add an in-text citation.

Response: Added-line 367.

Lines 311-314: Please re-phrase this sentence, as it is difficult to follow.

Response: This sentence has now been rephrased as requested–Lines 373-378.

Line 315: Are the “two groups” mentioned here the high and low PCT groups?

Response: Yes, now Corrected for clarity–Line 379.

Line 328: Please specify the “two values.”

Response: The two values were CRP and IL-6. We have now stated this as suggested–line 402. 

Line 355: State the number of patients (46) instead of “good number.”

Response: This has now been corrected.

There are many typographical errors, such as missing hyphens and inconsistent hyphenation, missing commas, missed spacing between words, missing articles before nouns, and missing or incorrect punctuation.

Response: We have reviewed the entire manuscript and corrected as relevant.

---

## [Decision Letter · Decision Letter 1]

14 Feb 2022

PONE-D-21-26044R1A cytokine panel and procalcitonin in COVID-19, a comparison between intensive care and non-intensive care patientsPLOS ONE

Dear Dr. Mazaheri,

Thank you for submitting your manuscript to PLOS ONE. After careful consideration, we feel that it has merit but does not fully meet PLOS ONE’s publication criteria as it currently stands. Therefore, we invite you to submit a revised version of the manuscript that addresses the points raised during the review process.

We look forward to receiving your revised manuscript.

Kind regards,

Dong Keon Yon, MD, FACAAI

Academic Editor

PLOS ONE

Journal Requirements:

Additional Editor Comments:

Firstly, I am apologize for the delay (your paper has been delayed due to changes in the editor). Thank you for submitting your manuscript to Plos One. The reviewers and I believe it is of potential value for our readers. Please address minor comments of the reviewer #2.

Please cite the top-tier papers.

#1. Yang JM, Koh HY, Moon SY, Yoo IK, Ha EK, You S, Kim SY, Yon DK, Lee SW. Allergic disorders and susceptibility to and severity of COVID-19: A nationwide cohort study. J Allergy Clin Immunol. 2020 Oct;146(4):790-798. doi: 10.1016/j.jaci.2020.08.008. Epub 2020 Aug 15. PMID: 32810517; PMCID: PMC7428784.

#2. Shin YH, Shin JI, Moon SY, Jin HY, Kim SY, Yang JM, Cho SH, Kim S, Lee M, Park Y, Kim MS, Won HH, Hong SH, Kronbichler A, Koyanagi A, Jacob L, Smith L, Lee KH, Suh DI, Lee SW, Yon DK. Autoimmune inflammatory rheumatic diseases and COVID-19 outcomes in South Korea: a nationwide cohort study. Lancet Rheumatol. 2021 Oct;3(10):e698-e706. doi: 10.1016/S2665-9913(21)00151-X. Epub 2021 Jun 18. PMID: 34179832; PMCID: PMC8213376.

Reviewers' comments:

Reviewer's Responses to Questions

**Comments to the Author**

1. If the authors have adequately addressed your comments raised in a previous round of review and you feel that this manuscript is now acceptable for publication, you may indicate that here to bypass the “Comments to the Author” section, enter your conflict of interest statement in the “Confidential to Editor” section, and submit your "Accept" recommendation.

Reviewer #2: (No Response)

2. Is the manuscript technically sound, and do the data support the conclusions?

Reviewer #2: Yes

3. Has the statistical analysis been performed appropriately and rigorously? 

Reviewer #2: I Don't Know

4. Have the authors made all data underlying the findings in their manuscript fully available?

Reviewer #2: Yes

5. Is the manuscript presented in an intelligible fashion and written in standard English?

Reviewer #2: No

6. Review Comments to the Author

Reviewer #2: The authors addressed most of the previous comments, which improved the structure and coherence of the manuscript. There are still clarifications needed and improved proofreading needed.

• There are numerous misspellings/typographical errors. Thorough proofreading of phrases is needed.

• The terminology of co-infection, superadded infection, superinfection needs to be clearer. If the intent is for the same meaning across the manuscript, there needs to be consistency with the appropriate term. The nuances of these terms may need to be defined in the manuscript if the intent is for a molecular-based definition.

• Lines 111-112: At the very least, cite the published literature concerning the cytokine panel creation. It is unclear if this particular cytokine panel is location-specific or more widely used.

• Line 149: Which co-morbidities were defined as absent from “healthy” volunteers?

• Methods: Which specific bacterial infections were included in the testing?

• Methods: There are discrepancies between the number of patients in certain groups and the numbers listed in Figure 1. Please review and correct the patient numbers.

• Discussion: The discussion states that there are few limitations, but that is not accurate. As a retrospective study, there are several inherent limitations. Also, healthy volunteers were not tested for COVID-19 (as explained in the Methods), but this is still another limitation due to lack of confirmation.

7. PLOS authors have the option to publish the peer review history of their article (what does this mean?). If published, this will include your full peer review and any attached files.

Reviewer #2: No

---

## [Author Response · Author response to Decision Letter 1]

23 Mar 2022

Comment: *Please cite the top-tier papers.

#1. Yang JM, Koh HY, Moon SY, Yoo IK, Ha EK, You S, Kim SY, Yon DK, Lee SW. Allergic disorders and susceptibility to and severity of COVID-19: A nationwide cohort study. J Allergy Clin Immunol. 2020 Oct;146(4):790-798. doi: 10.1016/j.jaci.2020.08.008. Epub 2020 Aug 15. PMID: 32810517; PMCID: PMC7428784. #2. Shin YH, Shin JI, Moon SY, Jin HY, Kim SY, Yang JM, Cho SH, Kim S, Lee M, Park Y, Kim MS, Won HH, Hong SH, Kronbichler A, Koyanagi A, Jacob L, Smith L, Lee KH, Suh DI, Lee SW, Yon DK. Autoimmune inflammatory rheumatic diseases and COVID-19 outcomes in South Korea: a nationwide cohort study. Lancet Rheumatol. 2021 Oct;3(10):e698-e706. doi: 10.1016/S2665-9913(21)00151-X. Epub 2021 Jun 18. PMID: 34179832; PMCID: PMC8213376.

Response: The above two references have now been included under ‘Discussion’ – reference number 19 and 20.

Comment: There are numerous misspellings/typographical errors. Thorough proofreading of phrases is needed.

Response: Thank you for highlighting this. We have reviewed the manuscript and made relevant corrections throughout. 

Comment: The terminology of co-infection, superadded infection, superinfection needs to be clearer. If the intent is for the same meaning across the manuscript, there needs to be consistency with the appropriate term. The nuances of these terms may need to be defined in the manuscript if the intent is for a molecular-based definition.

Response: Concurrent bacterial infection and co-infection is now replaced with 

super-infection throughout because all confirmed bacterial infection in this cohort, occurred ≥48 hours after admission to hospital for COVID-19. (Lines 170-172)

Comment: Lines 111-112: At the very least, cite the published literature concerning the cytokine panel creation. It is unclear if this particular cytokine panel is location-specific or more widely used.

Response: Two additional references cited (Line 112). In addition the following sentence is now included in the manuscript - ‘While IL-6 is widely being used in the UK, this panel is unique to King’s College Hospital’. (Lines 111-116)

Comment:Line 149: Which co-morbidities were defined as absent from “healthy” volunteers?

Response: The sentence has now been re-structured for better clarity - Lines 153-155.

 Comment:Methods: Which specific bacterial infections were included in the testing?

Response: We have now included the details of the culture results - Lines 310-317.

 Comment: Methods: There are discrepancies between the number of patients in certain groups and the numbers listed in Figure 1. Please review and correct the patient numbers.

Response: The patient numbers have been corrected as relevant - Line 138.

 Comment: Discussion: The discussion states that there are few limitations, but that is not accurate. As a retrospective study, there are several inherent limitations. Also, healthy volunteers were not tested for COVID-19 (as explained in the Methods), but this is still another limitation due to lack of confirmation.

Response: This has now been addressed – Lines 432-436.

Yours sincerely,

Tina Mazaheri

---

## [Editor Report · Decision Letter 2]

25 Mar 2022

A cytokine panel and procalcitonin in COVID-19, a comparison between intensive care and non-intensive care patients

PONE-D-21-26044R2

Dear Dr. Mazaheri,

We’re pleased to inform you that your manuscript has been judged scientifically suitable for publication and will be formally accepted for publication once it meets all outstanding technical requirements.

Kind regards,

Dong Keon Yon, MD, FACAAI

Academic Editor

PLOS ONE

Additional Editor Comments (optional):

I congratulate you on this mesmerizing paper.
---

## [Editor Report · Acceptance letter]

8 Apr 2022

PONE-D-21-26044R2 

A cytokine panel and procalcitonin in COVID-19, a comparison between intensive care and non-intensive care patients 

Dear Dr. Mazaheri:

I'm pleased to inform you that your manuscript has been deemed suitable for publication in PLOS ONE. Congratulations! Your manuscript is now with our production department. 

Kind regards, 

on behalf of

Dr. Dong Keon Yon 

Academic Editor

PLOS ONE